# Vector Quantized Representations for Efficient Hierarchical Delineation of Behavioral Repertoires

## Abstract

Understanding animal behaviors and their neural underpinnings requires precise kinematic measurements plus analytical methods to parse these continuous measurements into interpretable, organizational descriptions. Existing approaches, such as Markov models or clustering, can identify stereotyped behavioral motifs out of 2D or 3D keypoint-based data but are limited in their interpretability, computational efficiency, and/or ability to seamlessly integrate new measurements. Moreover, these methods lack the capacity for capturing the intrinsic hierarchy among identified behavioral motifs (e.g., "turning" $\rightarrow$ subtypes of left/right turning with varying angles), necessitating subjective post hoc annotations by human labelers for grouping. In this paper, we propose an end-to-end generative behavioral analysis approach that dissects continuous body movements into sequences of discrete latent variables using multi-level vector quantization. The discrete latent space naturally defines an interpretable behavioral repertoire composed of hierarchically organized motifs, where the top-level latent codes govern coarse behavioral categories (e.g., rearing, locomotion) and the bottom-level codes control finer-scale kinematics features defining category subtypes (e.g., lateral sidedness). Using 3D poses extracted from recordings of freely moving rodents (and humans), we show that the proposed framework faithfully supports standard behavioral analysis tasks while enabling new applications stemming from the discrete information bottleneck, including realistic synthesis of animal body movements and cross-species behavioral mapping.

## 1 Introduction

To understand the neural underpinnings of animal behavior, including the mechanisms underlying motor control, decision making, social interaction, and neurodegenerative diseases, we must be able to quantify and delineate the animal behavior repertoire, monitoring its changes over time and responses to experimental perturbations (Anderson & Perona, 2014). Recent advances in videography-based tracking of animal movements via 2D or 3D pose estimation support high-resolution measurements of rich body kinematics in freely behaving animals across a wide range of species, experimental contexts and laboratories (Mathis et al., 2018; Bala et al., 2020; Pereira et al., 2022; Dunn et al., 2021; Marshall et al., 2022). However, the expanded scale and resolution of these acquired data pose new challenges: how can informative behavioral descriptions be derived from raw kinematic measurements (Datta et al., 2019)?

An ideal behavioral analysis pipeline would be able to detect and characterize behavioral patterns reproducibly across replicates and between labs and interpretably over large-scale animal dataset collected over time (Gomez-Marin et al., 2014). Most existing methods build off the idea that behavior can be discretized into series of stereotyped, recurring action motifs (e.g., grooming, rearing, locomotion) (Brown et al., 2013; Berman et al., 2014; Wiltschko et al., 2015) that, in turn, can be subdivided into finer-grained behavioral subtypes (e.g., low and high rearing) forming a relational hierarchy (Berman et al., 2016; Luxem et al., 2022; Voloh et al., 2023). By assigning motif identities to continuous pose sequences, behaviors are quantitatively profiled in terms of motif occurrence frequencies and transition probabilities, which can be further processed to identify behavioral structures over longer timescales as recurring sequences and states (Marshall et al., 2021).

While discretization of the behavioral repertoire into motifs gives rise to more human-comprehensible representations, it remains challenging to effectively construct such fundamental units, interpret their composition, and, especially, examine how motif assignments potentially vary in measurements collected in different individuals and experimental setups (Datta et al., 2019). While clustering-based methods (Berman et al., 2014; Marshall et al., 2021; Luxem et al., 2022; Hsu & Yttri, 2021) are advantageous for distinguishing subtle kinematic details, they often propose hundreds of distinct motifs. Interpreting clustering results thus requires manual human review and annotation, ideally in consensus among multiple experts, to assign each cluster to a coarse behavioral class (e.g., grooming) and a fine label reflecting the underlying kinematics (e.g., grooming left versus right side of the body). Furthermore, the most common clustering approaches rely on t-SNE, which does not support incorporation of new data without training a separate re-embedding model or rerunning the entire analysis pipeline, which remains inefficient and affects reproducibility (Van der Maaten & Hinton, 2008). Others tackle the problem using hidden Markov models (HMMs), describing behaviors as a series of behavioral states representing distinct pose dynamics with stationary transition probabilities (Wiltschko et al., 2015; Weinreb et al., 2023). HMM-based methods better identify transition points in behavioral sequences, typically find a narrower set of behavioral classes ($n < 30$), but this formulation fundamentally constrains the range and resolution of kinematics captured in the detected motifs and could affect its generalization to new data. Furthermore, like with clustering, Markov approaches do not model the motif type/subtype hierarchy, requiring manual annotation and grouping as part of the analysis pipeline. Others have utilized variational autoencoders (VAE) to map pose sequences to hierarchical behavioral motifs without human-guided annotation, but through a disconnected, serial process consisting of fitting an HMM followed by hierarchical clustering of Markov transition probabilities (Luxem et al., 2022).

Inspired by recent advances in unsupervised representation learning, including the discretization of continuous modalities such as audio (Schneider et al., 2019; Baevski et al., 2019; 2020; Dhariwal et al., 2020; Van Niekerk et al., 2020), we introduce an end-to-end unsupervised behavioral mapping method that identifies hierarchically organized behavioral motifs from postural time-series. At the core is a VAE mapping postural dynamics to a finite-sized discrete embedding "codebook" via vector quantization (VQ). Upon completion of training, the learned codes compose an interpretable behavioral repertoire of distinct, stereotyped motor primitives that can be leveraged to both analyze and synthesize movements (Fig. 1a). To account for behavior type/subtype hierarchy, we incorporate a novel multi-level encoding scheme that separates generative factors into high-level components representing coarse behavioral categories and low-level components controlling varying movement dynamics, thus providing a natural hierarchical summarization of identified (sub-)behaviors without *post hoc* comparisons and annotations (Fig. 1b). These hierarchical representations allow us to recapitulate the behavioral identification and segmentation of existing techniques while better capturing and grouping finer-scale kinematic details. Representing continuous movements as sequences of decipherable codes also improves motion synthesis compared to recent SOTA techniques, and further enables cross-species style transfer and motif identification, which open the door to new lines of inquiry in neuroscience.

Quantitative behavioral analyses have become a major focus of computational neuroscience, whose collective goal is to understand and model brain-behavior relationships. In summary, our contributions are as follows:

- We introduce a novel VQ-VAE framework with multi-level encoding for identifying hierarchically organized behavioral motifs, simplifying and accelerating behavioral analysis pipelines, with enhanced sensitivity to fine-grained postural details.
- Using VQ-VAE codes, we improve the quality of motion synthesis relative to recent approaches.
- We establish a new framework for aligning motion sequences between unpaired datasets, even across species.

## 2 BACKGROUND AND RELATED WORK

**Animal behavioral analysis** characterizes the organization and evolution of behaviors over time, in a way that best support inquires into the underlying neural principles and mechanisms (Wiltschko et al., 2020; Willmore et al., 2022; Voloh et al., 2023; Mimica et al., 2023; Graziano & Aflalo,

2007). The most common behavioral measurements are 2D and 3D keypoint-based body poses extracted *post hoc* from images/videos (Mathis et al., 2018; Pereira et al., 2022; Dunn et al., 2021). As illustrated in the Introduction, decomposing continuous movement dynamics into sequences of discrete building blocks is necessary for describing and understanding complex behavioral patterns in a more human-comprehensible way. With no explicit boundaries with respect to transitions between behaviors, their definitions may diverge significantly depending on human empirical knowledge and underlying modeling assumptions. In addition to supervised classifers that detect predefined animal behaviors (Aksan et al., 2019; Arac et al., 2019; Segalin et al., 2021; Nilsson et al., 2020; Sun et al., 2021), established unsupervised methods have adopted Markov assumptions and optimized for maximal likelihood of behavioral observations conditioned on probabilistic transitions between hidden states (Wiltschko et al., 2015; Weinreb et al., 2023; Luxem et al., 2022; Costacurta et al., 2022). The representation of behaviors, however, may deviate significantly from the Markovian dynamics with time-invariant transition probabilities and limited forecasting horizon (Berman et al., 2016; Marshall et al., 2021). In contrast, clustering-based methods (Berman et al., 2014; Marshall et al., 2021; Hsu & Yttri, 2021; Kwon et al., 2023) ignore long-term temporal structures, only independently crafting spatio-temporal dynamics features within a short local window (e.g., 500 ms by Marshall et al.) at each timepoint. Our method is conceptually more aligned with clustering, as quantization relies on nearest neighbor queries given a Voronoi partition of the behavioral repertoire, but with better computational efficiency and interpretability. Moreover, each embedding vector in the codebook, as hidden states in HMMs, can directly decoded into representative movement trajectories.

**Discrete representation learning** with deep learning has progressed since vector-quantized variational autoencoder (VQ-VAE) (Van Den Oord et al., 2017) remarkably closed the performance gap with continuous VAEs. VQ-VAE has become a popular choice for generative modeling of continuous modalities, such as audio and human body motion (Dhariwal et al., 2020; Ng et al., 2022; Zhang et al., 2023), where symbolic representations (e.g., pitch, note) might not be the most proper options for learning or do not exist (e.g., unified, precise definitions of actions). The limited number of candidates within the codebooks also mitigate the drifting issue when directly regressing on continuous modalities (Aksan et al., 2019). We identified the concept of neural discrete representation learning align with, and naturally resolves existing issues in current animal behavioral analysis pipelines.

A **hierarchical organization** of the learned discrete representations is additionally offered by the proposed method. In the context of vector quantization, works have independently quantized partitions of the latent variables using multiple codebooks (product quantization) which increases the overall code quantity and improves downstream performance (Baevski et al., 2019). For image synthesis, Singh et al. (2019) used a hierarchical layout of codebooks with similar motivations as our proposed method to help reveal the hierarchy among instances. Li et al. (2019) showed a hierarchy of latent variables can be progressively learned in VAEs where each abstraction level captures different generative factors and therefore more interpretable representations.

Finally, our work is related to the task of **generative motion modeling**. Works have applied VAEs for abstracting lower-dimensional motion representations and captured the temporal dynamics using recurrent neural networks (Yan et al., 2018; Guo et al., 2020) and transformers (Petrovich et al., 2021; 2022; Tang & Matteson, 2021). Works (Guo et al., 2022; Zhang et al., 2023; Siyao et al., 2022) have also exploited VQ-VAE for decomposing continuous motion sequences into finite-sized set of decipherable codes, paired with autoregressive models to enable diverse, flexible and controllable motion generation, optionally conditioned from other discrete modalities such as textual descriptions and music. While not being the main focus of the work, the capacity of synthesizing realistic animal motions could empower standardization of motion representations and creation of synthetic datasets that help improve and benchmark behavioral recognition and segmentation methods.

## 3 METHODS

### 3.1 VQ-VAE FOR BEHAVIORAL MEASUREMENTS

We first describe the preliminaries for the vector quantized VAE and how we deploy it for parsing behavioral measurements (Fig. 1a). Given inputs x, we define an encoder network $q(\cdot)$ that parameterizes $q_\phi(z|x)$, the posterior distributions of latent variables $z$ given inputs x, and a decoder network parameterized by $\theta$ approximating the likelihood of real data $p_\theta(x|z)$. In our case, the prior distribution $p(z)$ is uniform over a set of embedding vectors $\{e_i \in \mathbb{R}^D\}, i \in \{1, \dots, K\}$, which forms a

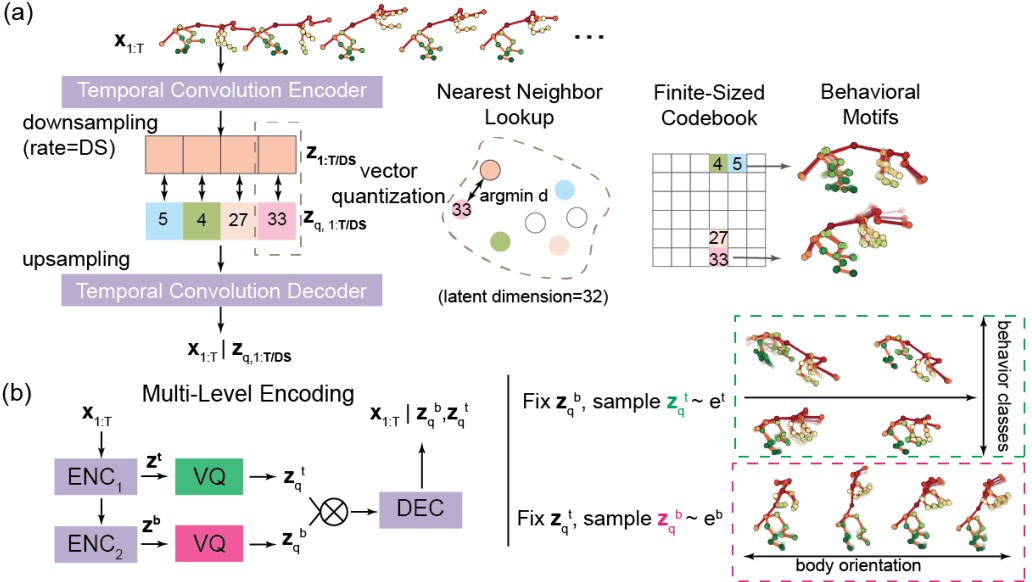

Figure 1: Method overview. (a) The temporal convolutional encoder compresses the animal posture time-series into a reduced set of latent variables, which is passed through the discrete bottleneck and mapped to the nearest embedding vector in the codebook. Each learned behavioral motif can be decoded into semantically meaningful kinematics. (b) We introduce a hierarchy of latent variables during the encoding process, where each level disentangles different generative factors from identified behaviors. Specifically, the top-level codes ($z_q^t$) emerge as coarse behavioral categories such as rearing and locomotion while the bottom-level codes ($z_q^b$) capture more subtle kinematic variations.

discrete embedding space $\mathcal{E} \in \mathbb{R}^{K^D}$ where $D$ is the dimensionality of each embedding vector. For simplicity, we refer to the discrete latent space as a dictionary of embedding vectors or codebook.

The posterior is defined as a one-hot categorical distribution,

$$q_\phi(\mathbf{z} = k|\mathbf{x}) = \begin{cases} 1 & \text{for } k = \mathrm{argmin}_i \|z - e_i\|_2 \\ 0 & \text{otherwise} \end{cases} \tag{1}$$

which retrieves the nearest neighbor of the low-dimensional encoder outputs $z$ from the codebook. The vector quantized representations can then be expressed as

$$z_q(\mathbf{x}) = \mathrm{VQ}(z) = \mathrm{e}_k \text{ where } k = \arg\min_i \|z - \mathrm{e}_i\|_2 \tag{2}$$

Given $p(z)$ is uniformly distributed over $K$ embedding vectors, the associated KL divergence (if view as a variational autoencoder) $D_{KL}(q_\phi(\mathbf{z}|\mathbf{x})||p(z)) = \sum_{k=1}^{K} q_\phi(\mathbf{z} = k|\mathbf{x}) \log \frac{q_\phi(\mathbf{z}=k|\mathbf{x})}{p(z)}$ reduces to a constant of $\log K$ and can be ignored in the training objective.

The nearest-neighbor operations are not differentiable and thus there exists no gradients through the codebook. For optimization, we directly copy the gradients from the decoder to the encoder. Combining with the negative log-likelihood of the reconstruction, the learning objective is given by

$$\mathcal{L} = \log p(x|z_q(x)) + \|\mathrm{sg}[z_e(\mathbf{x})] - e\|_2^2 + \alpha\|z_e(x) - \mathrm{sg}[e]\|_2^2$$
$$= \mathcal{L}_{\text{reconstruction}} + \mathcal{L}_{\text{embedding}} + \alpha\mathcal{L}_{\text{commitment}} \tag{3}$$

where sg represents the stopgradient operator with zero gradients. Embedding vectors are zero initialized and updated through exponential moving average (Zhang et al., 2023). Last, the log-likelihood of the model is $\log p(x) = \log \sum_k p(x|z_k)p(z_k)$ with the approximation $\log p(x) \approx \log p(x|z_q(x))p(z_q(x))$, assuming that $z_q(x)$ should be zero for all $z_i \neq z_q$.

For our applications, the inputs x are temporal sequences of body poses $\mathbf{X} \in \mathbb{R}^{T \times (J \times D_J)}$, where $T$ is the number of time points, $J$ is the number of keypoints in the specific skeleton representation, and

$D_J$ is the dimensionality of the coordinate system (e.g., $D_J = 3$ for a Cartesian 3D system used in motion capture). The encoder and decoder are designed to perform downsampling and upsampling along the time dimension, such that once the model full converges, the learned codebook comprises a collection of short-term motor behavioral motifs instead of stationary views of body postures.

## 3.2 HIERARCHICAL ENCODING

While the vanilla VQ-VAE faithfully maps the continuous measurements of behaviors into a discrete set of interpretable behavioral motifs, the set is not compactly organized, with subtypes of the same behavior but variable kinematics (e.g., motifs of rat rearing up with its head tilted in different angles) often split into distinct and disjoint codes, as is common with existing behavioral analysis frameworks (see latent space comparisons between Fig. A5 and A.6). Hyper-fragmentation of the behavioral space can be mitigated by further mapping codes into coarse behavioral classes via simple clustering (Berman et al., 2014) or, for VQ-VAEs, promoting embedding vector similarity across temporal segments (Kamper & van Niekerk, 2020), but these *post hoc* approaches have limited ability to quantitatively characterize the nature and sources of variability. Here we propose a specific instance of VQ-VAE that disentangles behavioral variability components by introducing a hierarchy of latent variables (Fig. 1b). We decompose the discrete latent variables $z$ into two disjoint sets $z^b$ and $z^t$, respectively encoded and quantized by separate codebooks of size $K^b$ and $K^t$ at the bottom and top abstraction level of the encoding process. The prior distribution $p(z) = p(z^b)p(z^t)$ remains uniform over $N * M$ possible embedding vectors. We then approximate the posterior as

$$q_\phi(z|x) = q_\phi(z^b, z^t|x) = q(z^t|x)q(z^b|z^t, x) = q(z^t|x)q(z^b|z^t) \tag{4}$$

and optimize the previously describe objective in Section 3.1. This formulation is inspired by product, or multi-group, quantization (Jegou et al., 2010; Baevski et al., 2019), which reduces single embedding vectors into multiple groups of sub-vectors for computational and storage efficiency.

## 3.3 DATASETS & IMPLEMENTATION

**Animals.** Our primary focus was on lab rodents. We used two mouse 2D pose datasets (Keypoint-MoSeq2D Weinreb et al. (2023) and CalMS21 Sun et al. (2021)), one mouse 3D pose dataset (Li et al., 2023), and one rat 3D pose dataset "LE" of 30-minute behavioral recordings of 12 freely behaving Long Evans ("LE") rats. The "LE" dataset contains more tracked per-frame 3D keypoints (n=23) than 3D mouse dataset (n=20). A subset of the "LE" recordings were made after administration of amphetamine, allowing us to examine the ability of our method to detect drug-induced changes in behavior. See Section. A.1 for further details.

**Human.** To evaluate the cross-species behavioral identification, we used a curated version (Petrovich et al., 2021) of the UESTC RGB-D dataset (Ji et al., 2018), consisting of 13350 3D pose sequences across 40 human action categories.

For all datasets, we canonicalized all poses to their frontal view and remove all global translations and rotations. For all models, we set the codebook sizes to be 8 and 16 for the top and bottom layer, with embedding dimension of 32. We set the encoder downsamping rate to be 16 and thus each learned code spans over a minimum of 320 ms in the sequence. We used AdamW optimizer with $\beta_1 = 0.9$ and $\beta_2 = 0.99$ and performed cosine annealing with an initial learning rate of 0.0001 and for 450 epochs. Empirically, we set $\alpha = 0.02$ as the weight of the commitment loss.

## 4 RESULTS

Discrete representations established by the VQ-VAE can serve as foundations for animal behavioral analysis. In this section, we show how our method not only simplifies the traditional neuroscience behavioral analysis pipelines but also enables a series of novel applications for data unification and comparative, cross-species investigation.

## 4.1 DISCRETE REPRESENTATIONS ENABLE COMPREHENSIVE KINEMATIC PROFILING

We begin our discussion by qualitatively and quantitatively examining the behavioral repertoire characterized by the discrete codebooks. We retrieved the pose dynamics associated with each code's

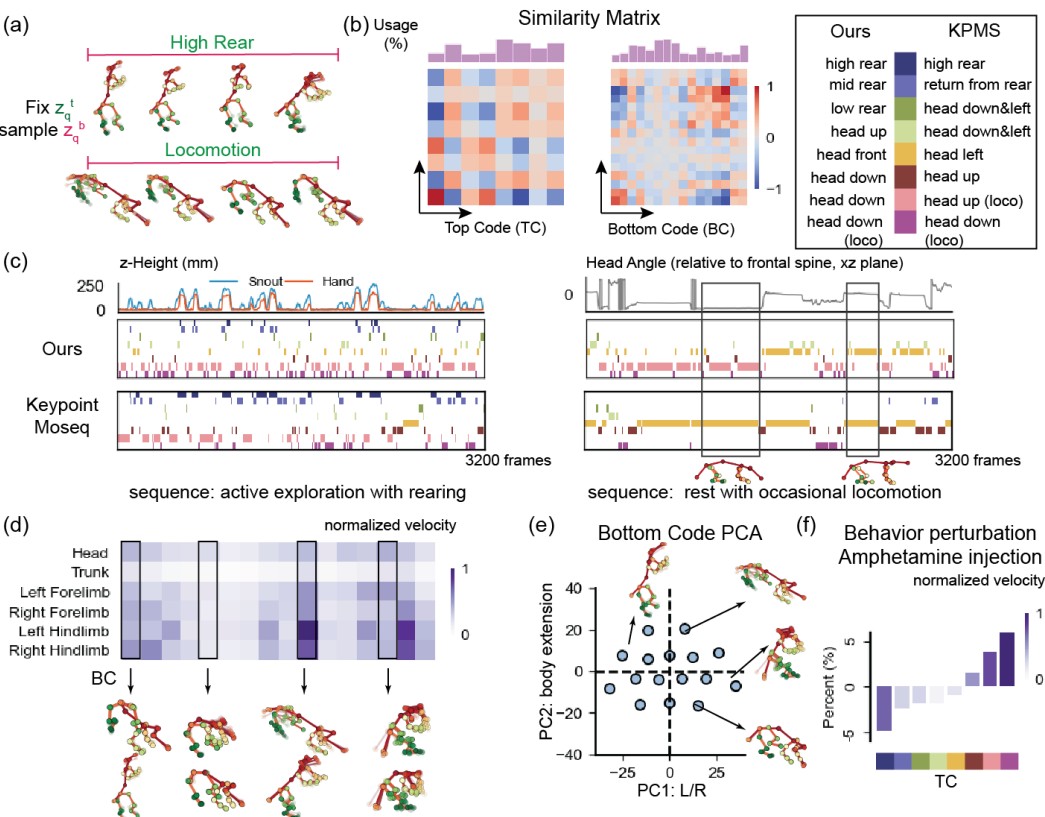

Figure 2: Kinematic profiling using VQ codes. (a) Hierarchy among the identified codes. (b) Pairwise cosine similarities of embeddings vectors. Similarity values are normalized to [-1, 1]. (c) Comparison of ethograms estimated by TCs from the proposed methods and by states of Keypoint-Moseq (KPMS). TCs were sorted by similarities to the first code computed in (b). (d) Perturbations in body part velocities across different BCs. (e) PCA space of the BCs. (f) Behavioral perturbation of rats after amphetamine injections.

embedding vector by feeding them into the probabilistic decoder. When reviewing different top and bottom-level code combinations (Fig. A5), we observed disentangled representations of behavioral features across the two codebooks: the smaller collection of top-level codes (TC) governed the basic coarse behavioral categories and the more numerous bottom-level codes (BC) accounted for variation within each each coarse category, forming subtypes of behaviors. These distinctions were reflected in the pairwise cosine similarity of code embedding vectors (Fig. 2b).

We generated "ethograms" from inferred TCs, which reflected the usage of common high-level motifs without *post hoc* grouping or clustering of the larger discrete code space, and qualitatively compared them to results from Keypoint-MoSeq (KPMS) (Weinreb et al., 2023), a recent extension of the well-established MoSeq autoregressive HMM from depth imaging to keypoint data. We specified the number of KPMS states to match the number of TCs (n = 8). Visualizations of the complete VQ-VAE codebook and KPMS syllables are in SectionA.2. Manually sorting KPMS syllables to match VQ codes revealed substantial qualitative similarities in behavioral segmentation, although differences were observed when comparing each method's behavioral category assignments to kinematic parameters derived from ground truth postures. We observed that the first three TCs were particularly attuned to different types/stages of rearing events, which were distinguished by deviations in the z-height of the animal's snout and hand (Fig.2c, Left). When examining a sequence during which the animal alternated between resting and locomotion, VQ codes identified a transition between behavioral categories not observed in KPMS. Further review revealed that the transitions in VQ codes reflected a stable change in the animal's resting head angle (Fig. 2c, Right, black boxed

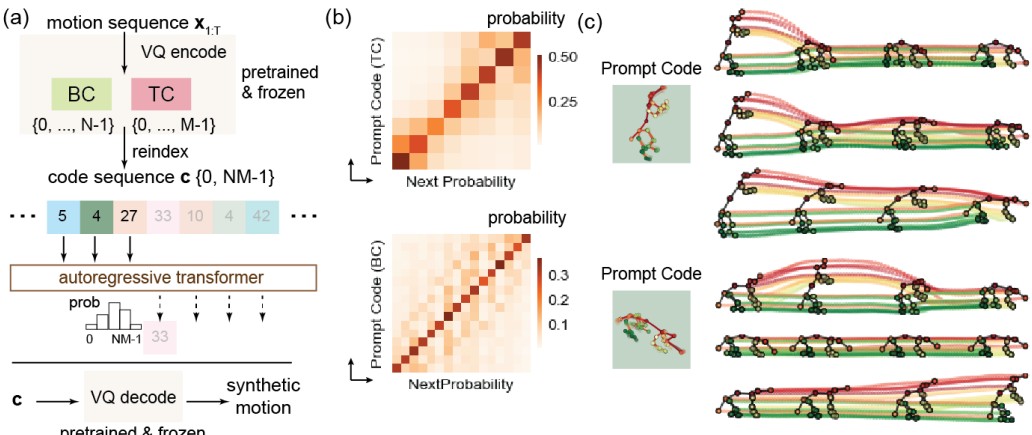

Figure 3: Autoregressive generative modeling on code sequences. (a) Schematic illustrating how continuous motions are converted into discrete code sequences using a pretrained hierarchical VQ-VAE model, on top of which autoregressive models can be trained, and how generated code sequences can be decoded to motion. (b) Probabilities for the next code (top, bottom) with one single prompt code. (c) Qualitative examples of synthetic motion sequences.

regions). These results suggest that the VQ TCs can recapitulate the motif identification of existing techniques while capturing finer scale kinematic details.

Fig. 2d summarizes how BCs modulate body kinematics within the coarse behavioral categories captured by TCs. Principal component analysis of the BC embedding vectors revealed two primary axes of kinematic variation, with PC1 associated with the orientation of the animal (left/right), and PC2 representing body extension/compactness (Fig. 2e). To quantify the degree of BC/TC disentanglement, we quantified the robustness of TCs to synthetic perturbations of pose sequences. We found that our hierarchical formulation better separated behavioral components than a single-level codebook and, further, were able to identify optimal BC and TC sizes (Appendix A.5).

Last, to test the ability of our method to identify behavioral perturbations, we used our trained model to analyze recordings with and without amphetamine administration. As shown in Fig. 2f, VQ codes indicated upregulation of fast locomotion and downregulation of rearing, on average.

Our method is also significantly faster at inference. Excluding time for data preprocessing and initialization, it took approximately 1.89 seconds on a single GPU to assign codes to an entire dataset of 30 recordings (63 ms / recording). KPMS, while fast overall, was slow compared to the VQ-VAE, taking 4 seconds for a single recording.

## 4.2 DISCRETE REPRESENTATIONS ENABLE REALISTIC MOTION SYNTHESIS

Motion synthesis and forecasting are long-standing problems in human motion modeling research, but realistic synthesis of animal motion has not yet been explored extensively (Liu et al., 2019; 2021). Here, we show that our learned codebooks enable conditioned and unconditioned synthesis of realistic motion sequences, significantly outperforming alternative approaches (Guo et al., 2020; Petrovich et al., 2021). The motion synthesis model itself learns and generates behavioral patterns only on top of VQ-VAE code sequences, which can then be decoded into pose time series by the VQ-VAE *post hoc*, both illustrating the sufficiency of discrete behavioral representations and providing a compression strategy to help motion synthesis scale.

To generate long motion sequences probabilistically, we trained an autoregressive transformer (4 layers, 4 attention heads, embedding dimension of 128) using a cross-entropy loss. For the VQ-VAE, code sequences were extracted from a pretrained, frozen encoder (c.f. Section 4.1), with the $M = 16$ bottom- and $N = 8$ top-level codes re-indexed into one unified code set of size $M * N = 128$. The transformer uses causal attention and is trained to predict the next code in the sequence, and any generated code can be decoded into full-body motion using the pretrained VQ-VAE decoder (Fig.

Table 1: Quantitative evaluation results for motion synthesis.

| Method | FID↓ | Diversity → |
|---|---|---|
| Real* | $0.129^{\pm 0.069}$ | $7.145^{\pm 0.45}$ |
| Code Reconstruction (DS=16) | $0.137^{\pm 0.072}$ | $7.132^{\pm 0.38}$ |
| Fully Connected | $46.77^{\pm 1.40}$ | $2.488^{\pm 0.13}$ |
| GRU | $10.79^{\pm 1.27}$ | $5.117^{\pm 1.17}$ |
| Action2Motion (conditional GRU) | $9.769^{\pm 0.73}$ | $\mathbf{7.189^{\pm 0.48}}$ |
| ACTOR Transformer (Petrovich et al., 2021) | $5.059^{\pm 0.84}$ | $5.568^{\pm 0.43}$ |
| MotionDiffuse (Zhang et al., 2022) | $2.839^{\pm 0.51}$ | $6.448^{\pm 0.56}$ |
| Code Transformer (VQ, 128 codes, DS=16) | $3.187^{\pm 0.33}$ | $6.441^{\pm 0.36}$ |
| Code Transformer (VQ, 64 codes, DS=16) | $2.843^{\pm 0.19}$ | $6.353^{\pm 0.38}$ |
| Code Transformer (Ours, DS=8) | $2.792^{\pm 0.60}$ | $6.620^{\pm 0.40}$ |
| Code Transformer (Ours, DS=16) | $\mathbf{1.717^{\pm 0.14}}$ | $7.211^{\pm 0.27}$ |
| Code Transformer (Ours, DS=32) | $2.491^{\pm 0.22}$ | $6.555^{\pm 0.30}$ |

4a). If desired, generated motion sequences can be conditioned on a specific starting code. We report learned code transition probabilities in Fig. 4b.

Fig. 4c shows qualitative examples of different sequences generated from single starting code (video examples with longer duration in Supplementary materials). To benchmark our model, we quantified synthesis quality over 64-frame movement sequences against several alternative VAE encoder/decoder architectures and state-of-the-art methods. Following previous works, we quantified the fidelity (Fréchet inception distance relative to real data) and diversity of 2000 generated sequences (Guo et al., 2020) from action features extracted by a pretrained convolutional denoising autoencoder (Tao et al., 2022). All statistics were averaged over 20 trials with different random seeds. Motion synthesis by discrete codes produced pose sequences that were significantly more similar to real data 1. Qualitatively, we found that the discrete code model clearly improved known problems with over-smoothing and drifting characteristic of generative motion models (Aksan et al., 2019; Ng et al., 2022), like because the VQ-VAE sampled and assembles motion sequences only from a set of prototypical movement units.

### 4.3 DISCRETE REPRESENTATIONS ENABLE CROSS-SPECIES BEHAVIOR MAPPING

Our VQ-based method can also learn joint behavioral latent representations of data from different sources (same species, different keypoint configurations) and even vastly different species (Fig. 4a). We extend the base formulation by incorporating modality/species-specific encoding and decoding units, as parameterized by single fully connected layers. For unpaired data $\boldsymbol{x}^{(1)}$ and $\boldsymbol{x}^{(2)}$ with different acquisition sources (e.g., rat and mice, rodents and human), we want to maximize the likelihood of reconstructing motion sequences for the mixture of data points

$$\max_{\phi,\theta,\phi_1,\phi_2,\theta_1,\theta_2} \mathbb{E}_{\boldsymbol{x}^{(1)}}\left[\mathbb{E}_{\mathbf{z}\sim \boldsymbol{q}^1_{\phi_1}(\boldsymbol{q}_\phi(\boldsymbol{x}))}\log p^1_{\theta_1}\left(p_\theta(\boldsymbol{x})|\boldsymbol{z}\right)\right] + \mathbb{E}_{\boldsymbol{x}^{(2)}}\left[\mathbb{E}_{\mathbf{z}\sim \boldsymbol{q}^2_{\phi_2}(\boldsymbol{q}_\phi(\boldsymbol{x}))}\log p^2_{\theta_2}\left(p_\theta(\boldsymbol{x})|\boldsymbol{z}\right)\right]$$

(5)

while subject to the VQ objectives as previously described. Our aim is to learn a unified high-level representation space abstracting the motion dynamics through the compact discrete information bottleneck, invariant to the low-level details embedded in different body profiles. The modality/species-specific encoding and decoding layers form mappings connecting the common latent space.

We first jointly embedded two rodent species (mouse and rat) with, we hypothesized, particularly aligned behavioral patterns and body plans. Qualitative examination of the embedding space revealed substantial similarities in movement dynamics across the decoded motifs for each species (Fig. 4b; Fig. A8-9). This setup can also be used to convert motion sequences from one species to the other using the joint latent space as the medium (Fig. 4c).

We then combined rats with the bipedal humans, which significantly deviate in their body plans. The resulting behavioral motifs suggested that the VQ-VAE could learn body-agnostic similarities in movements. Specifically, rat rearing behaviors were consistently associated with human up-down

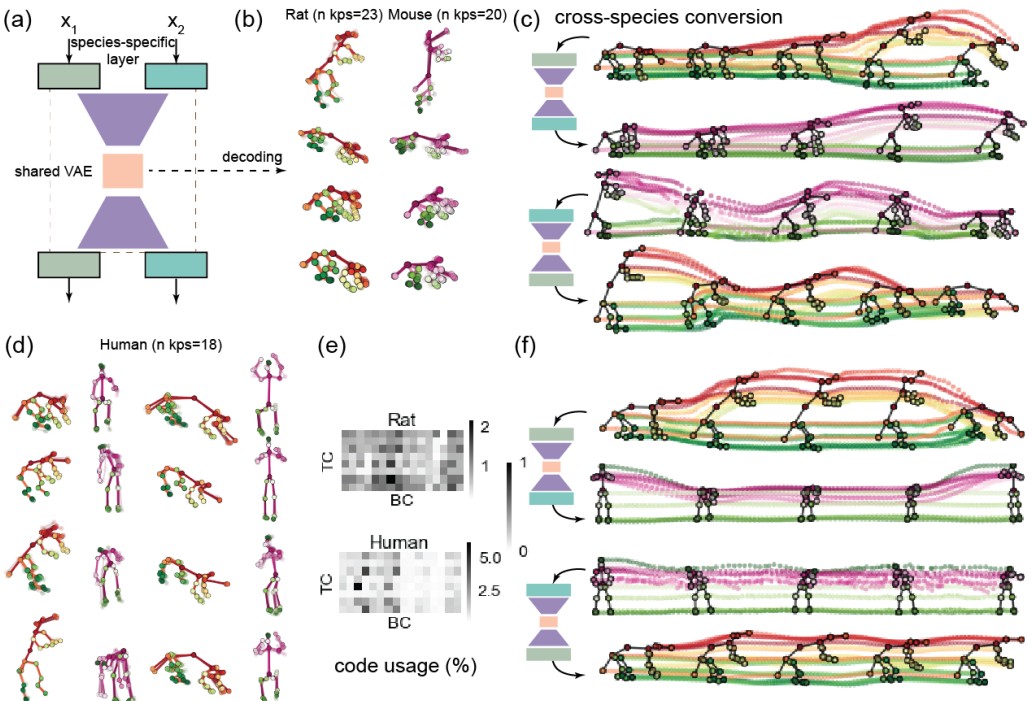

Figure 4: Cross-species behavioral mapping. (a) Schematic of the cross-species model. The dashed region corresponds to architecture previously described in Fig .1. The species-specific layers are single-layered fully connected modules. (b) Qualitative visualization of codes from joint encoding of rat and mouse data. (c) Conversion across species from interleaving encoders and decoders. (d, f) Qualitative results from rat-human joint embeddings. (e) Code usage in rat and human dataset.

movements of the torso (bending over), and rat locomotion was associated with human limb movements (Fig. 4d). While code usages were mostly overlapping and non-segregated, human data were represented by fewer codes on average. One explanation is that the UESTC human dataset contains a relatively limited set of action classes in comparison to the naturalistic behavioral repertoire.

Lastly, a shared discrete representation can also support co-analysis of pose data collected from the same species but with different annotated keypoint configurations. Differences in keypoint placement are common across labs, making it challenging for the community to combine datasets and reproduce results. When we jointly embedded sequences from both the KPMS-2D and CalMS21 datasets, we found that codes represented shared motifs reflecting the movement patterns common to and detectable from both keypoint sets (Fig. 11-10).

## 5 DISCUSSION AND CONCLUSION

In this paper, we introduce an efficient and versatile framework capable of parsing keypoint-based behavioral data into a hierarchically organized, discrete latent space of interpretable behavioral motifs. The discrete bottleneck makes efficient use of the latent space to capture salient kinematic features. When paired with autoregressive models, the learned discrete codes support high-quality generative synthesis of behavioral data and open the door to future comparative studies of behavior and new strategies for data standardization across labs. For future works, our model could also be extended to align behavioral representations to other modalities, such as vocalizations and neural activity. One limitation of our method is that, while longer-timescale relationships and dependencies can be modeled after training the VQ-VAE, the VQ codes themselves remain at a constant timescale. Imposing hierarchical structure over the time, in the spirit of Razavi et al. (2019) and Dhariwal et al. (2020), or using variable-length segments (Cuervo et al., 2022), could potentially improve the current framework and able more diverse applications.

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

# A APPENDIX

## A.1 DATASET DOCUMENTATION

We describe the datasets used in our experiments in detail as supplement to Sec.3.3 in the main text.

**LE dataset**. The LE dataset consists of 3D poses from 60 30-minute recordings of 6 male Long Evans rats. Eight of the recordings were made after administering 1.25 mg/kg amphetamine. All recordings were made with 6 synchronized, calibrated color cameras at 50 fps. We adapted an open-source animal pose estimation tool to obtain the 3D poses of all recordings. Each 3D pose consists of 23 keypoints covering the animal's head, trunk, forelimbs and hindlimbs.

The amphetamine subset was only deployed for the analysis in Fig. 2f and we used the first subset of recordings for model training, with six recordings withheld for validation.

**3D Mouse dataset** (Li et al., 2023). The dataset contains 2 3-minute recordings and 3 3-hour recordings with five mice, with a total of 1.1M frames. This public dataset is not annotated for all frames, so we acquired 20-keypoint 3D poses following the similar procedures as the LE. We downsampled the original recordings (100 fps) by 2.

## A.2  SINGLE-SPECIES BEHAVIORAL MAPPING

In this section, we show the following results

- Figure 5: visualization of the embedding space learned in our proposed method.
- Figure 6: visualization of the embedding space learned in the vanilla VQ-VAE.
- Figure 7: visualization of the behavioral modules identified by Keypoint-Moseq.

### A.3 CROSS-SPECIES BEHAVIORAL MAPPING

In this section, we show the following results

- Figure 8, Figure 9: visualization of complete latent space in the rat-mouse joint embedding experiment.
- Figure 11, Figure 10: visualization of the latent space learned from jointly embedding the two 2D datasets using VQ-VAE. We did not deploy the hierarchical version here given that movement patterns abstracted by these overhead 2D postures with coarse keypoint annotations are very limited and do not form a hierarchy.

### A.4 ADDITIONAL QUANTITATIVE COMPARISONS

We additionally performed quantitative analyses on the behavioral segmentation and identification performance of our proposed method, KPMS, as well as VQ-VAE with no hierarchy. We use the following metrics:

- Perplexity: a common information metric used for sequence analyses (e.g., language models), given by $PP(x) = 2^{H(x)}$, where H(x) is the entropy of code usage. For codebook/motif learning, lower perplexity implies possibilities for codebook collapse, i.e., a smaller number of active embedding vectors in the codebook. Higher perplexity thus implies more diverse sampling of the behavioral repertoire, i.e. that fine-grained differences in behaviors are detected and assigned to separate codes rather than being subsumed by a smaller number of coarse codes. For these comparisons, we fixed the codebook size (for quantization) and number of KPMS hidden states to be $n = 128$. For KPMS, we also tested multiple $\kappa$ parameter values, which controls state transition frequency.

  From Supp. Table. 2, despite tuning of the hyperparameter $\kappa$, KPMS yielded perplexity significantly lower than both a non-hierarchical and hierarchical VQ-VAEs, indicating that these VQ methods better capture fine-grained behaviors. At the same time, our hierarchical method not only captures these fine-grained behaviors but also groups them automatically into coarser categories, offering multiple scales of description in one shot.

- Jensen Shannon Divergence (JS Divergence): We computed distributions of code/syllable usage for different subtypes under the same coarse behavior category (e.g., high rearing versus low rearing) and assess the method's capacity for distinguishing subtypes via JS divergence (the higher the better).

  Due to the lack of continuous ground truth behavior annotations, we applied a clustering-based behavioral mapping method (Marshall et al., 2021). Each cluster was independently annotated with a detailed textual description (e.g., "low rear; upper body swaying"), as well as a coarse behavioral class label ("rearing") by two human annotators after viewing movies of randomly sampled movements. Any clusters with disagreement in annotators were reviewed until consensus. Different subtypes (e.g., "low rear", "high rear") were later discovered from the fine-grained descriptions.

  From the results summarized in Supp. Table. 3, our method is more capable of detecting subtle differences in kinematics than KPMS, especially in behavioral subtypes with salient postural changes. Even with only 8 and 16 codes, the performance of our method is on par with the non-hierarchical VQ-VAE, which contains 128 distinct codes.

Table 2: Code/Syllable Usage

|  | KPMS ($\kappa = 10^3$) | KPMS ($\kappa = 10^5$) | KPMS ($\kappa = 10^7$) | VQ-VAE (no hierarchy) | Ours (TC=4, BC=32) | Ours (TC=8, BC=16) |
|---|---|---|---|---|---|---|
| Perplexity ↑ | 16.63 | 15.17 | 14.94 | 56.23 | 57.38 | **57.87** |
| ≥ 1% usage | 15.63% | 13.28% | 12.5% | 17.19% | 21.88% | **23.44%** |

Table 3: JS Divergence in Behavioral Subtypes

| Coarse Class | Rearing | Idle | Locomotion |
|---|---|---|---|
| Subtypes | High Rear-Low Rear | Head Up-Head Down | Slow-Fast |
| KPMS ($\kappa = 10^3$) | 0.389 | 0.222 | 0.213 |
| KPMS ($\kappa = 10^5$) | 0.360 | 0.175 | 0.219 |
| KPMS ($\kappa = 10^7$) | 0.303 | 0.155 | 0.282 |
| VQ-VAE (no hierarchy) | 0.748 | 0.606 | 0.356 |
| Ours (TC=4, BC=32) | 0.611 | 0.557 | 0.370 |
| Ours (TC=8, BC=16) | 0.610 | 0.561 | 0.340 |
| Ours (TC=4, BC=16) | 0.578 | 0.435 | 0.325 |
| Ours (TC=8, BC=12) | 0.631 | 0.545 | 0.278 |
| Ours (TC=8, BC=32) | 0.700 | 0.580 | 0.366 |

## A.5 WHEN DOES BEHAVIORAL DISENTANGLEMENT FAIL

We leveraged a simple purity metric to quantify the degree of disentanglement in the identified codes. We augmented the original data by mirroring the bodies in the lateral plane. Both the original and the augmented data were encoded by different pretrained models. We calculated the average percentage of samples with a mismatch in their code assignments in each level. For example, if one data sample encoded as top code 4 is assigned to a different top code after augmentation, one mismatch is counted.

For this set of experiments, we fix the max number of possible codes to be 144 and vary their distributions in the top and bottom codebooks. Additionally, we compare to a flattened version of the proposed method based on vector quantization, where the latent variables are split equally and respectively encoded by two VQ codebooks. We report the highest purity across all levels.

From Table.4, the highest purity was achieved by the combination of top codebook of size 8 and bottom codebook of size 18 (denoted as (8, 18) for clarity). Its converse (18, 8), however, does not have equivalent performance, which reveals asymmetry in the generative factors that are captured in each encoding level. Notice that the extreme distribution (36, 4), although the metric stays low, the number of possible candidates is very small (4). The multi-group vector quantization does not achieve good disentanglement performance across identified motifs, suggesting that the hierarchy of latent variables introduced in our method is the key to disentanglement.

Table 4: Code Purity (N = 144) $\downarrow$

| | Top Codebook Size | | | | | | Multi-Group |
|---|---|---|---|---|---|---|---|
| | 4 | 8 | 12 | 18 | 24 | 36 | 8 |
| purity | 0.623 | 0.401 | 0.547 | 0.684 | 0.787 | 0.479 | 0.593 |

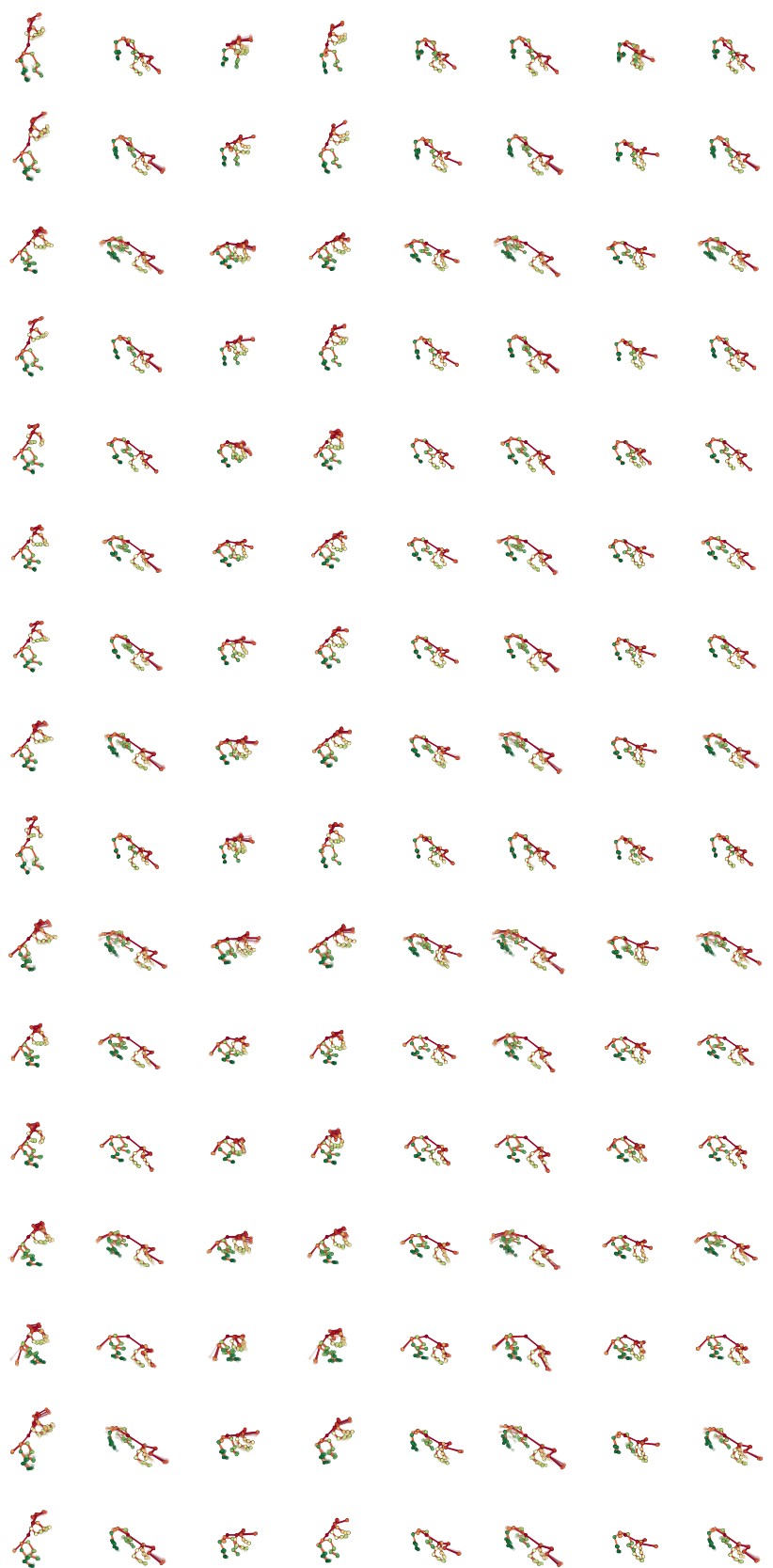

Figure 5: Visualization of the LE embedding space learned by our proposed hierarchical method. The row corresponds to the top codes (coarse behavioral categories) and the column corresponds to different bottom codes (subtypes).

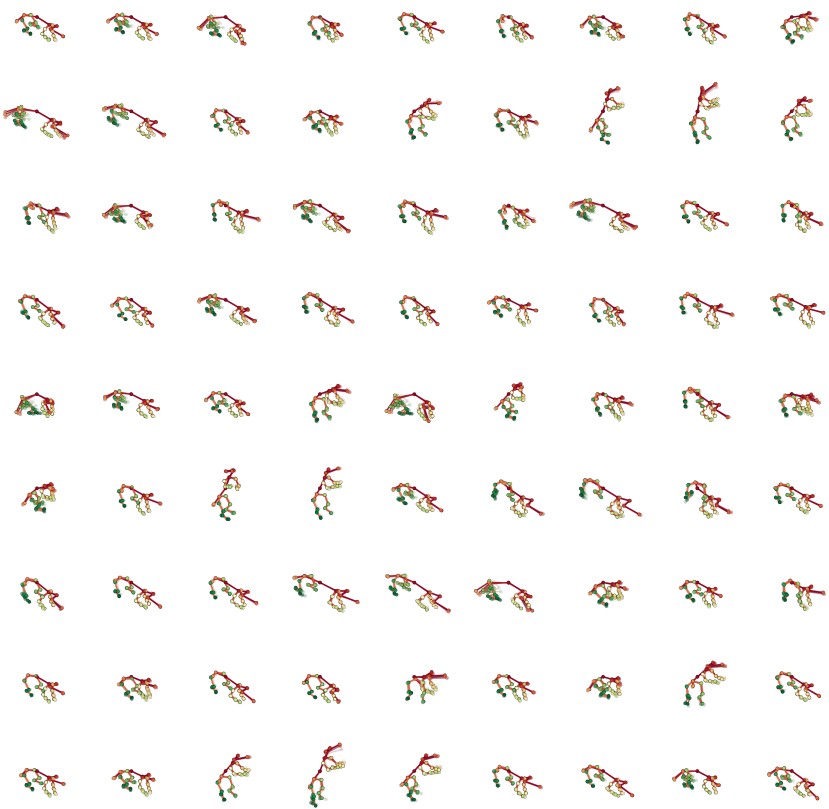

Figure 6: Visualization of the LE embedding space learned by vanilla VQ-VAE. Given the space limitation, we only show a subset of codes.

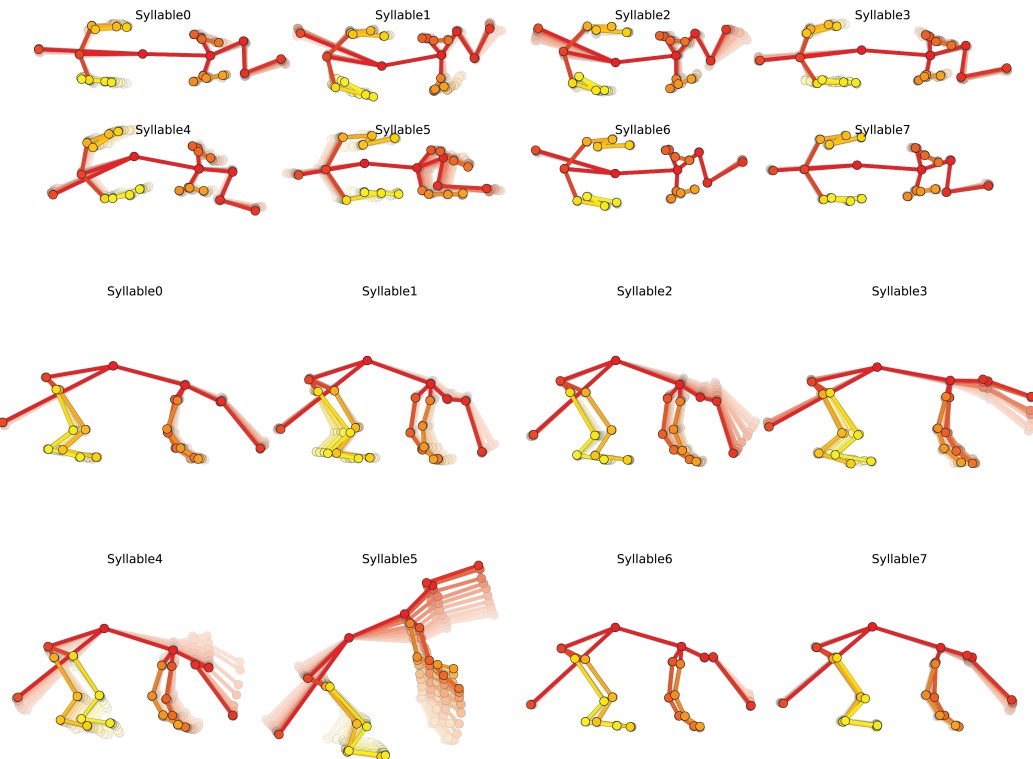

Figure 7: Visualization of behavioral motifs from Keypoint-Moseq.

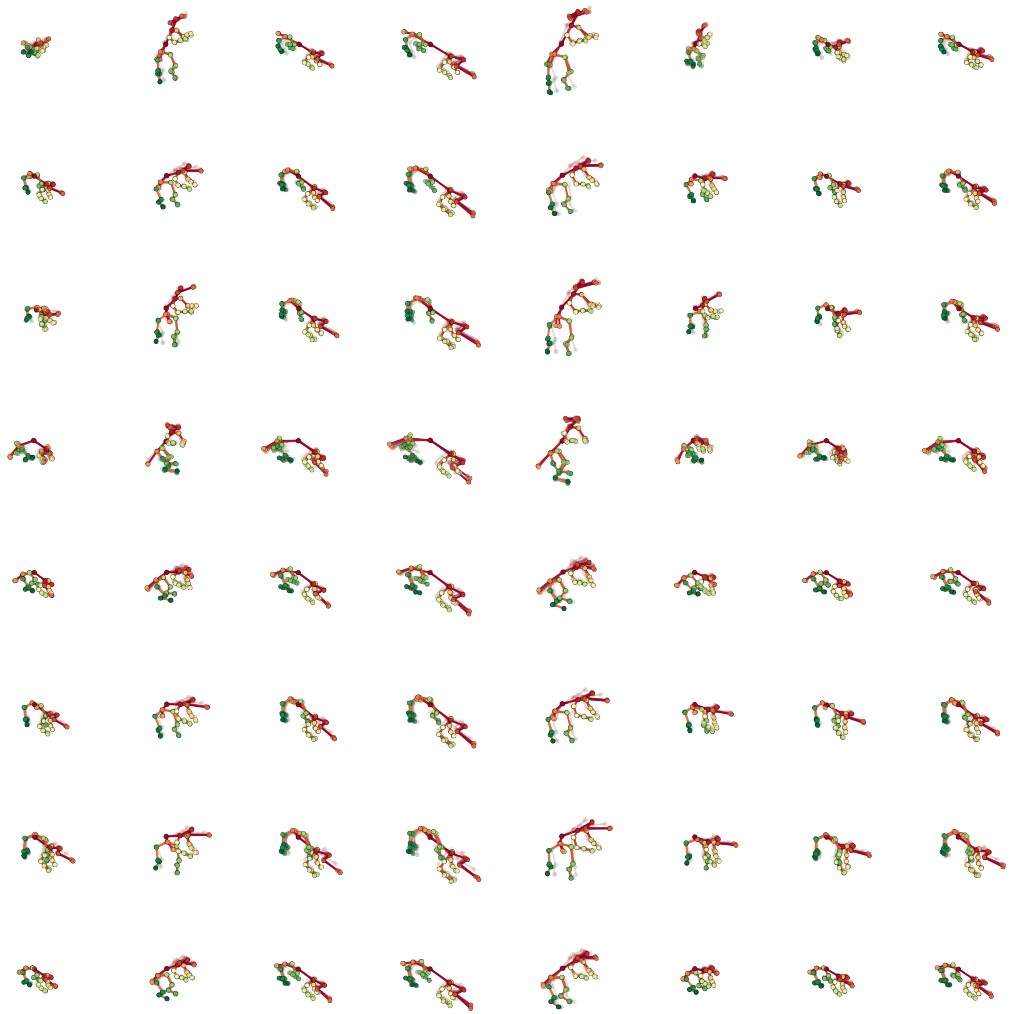

Figure 8: Visualization of the rat-mouse joint embedding space decoded as rats. For the limitation of space, only the first eight bottom codes are shown.

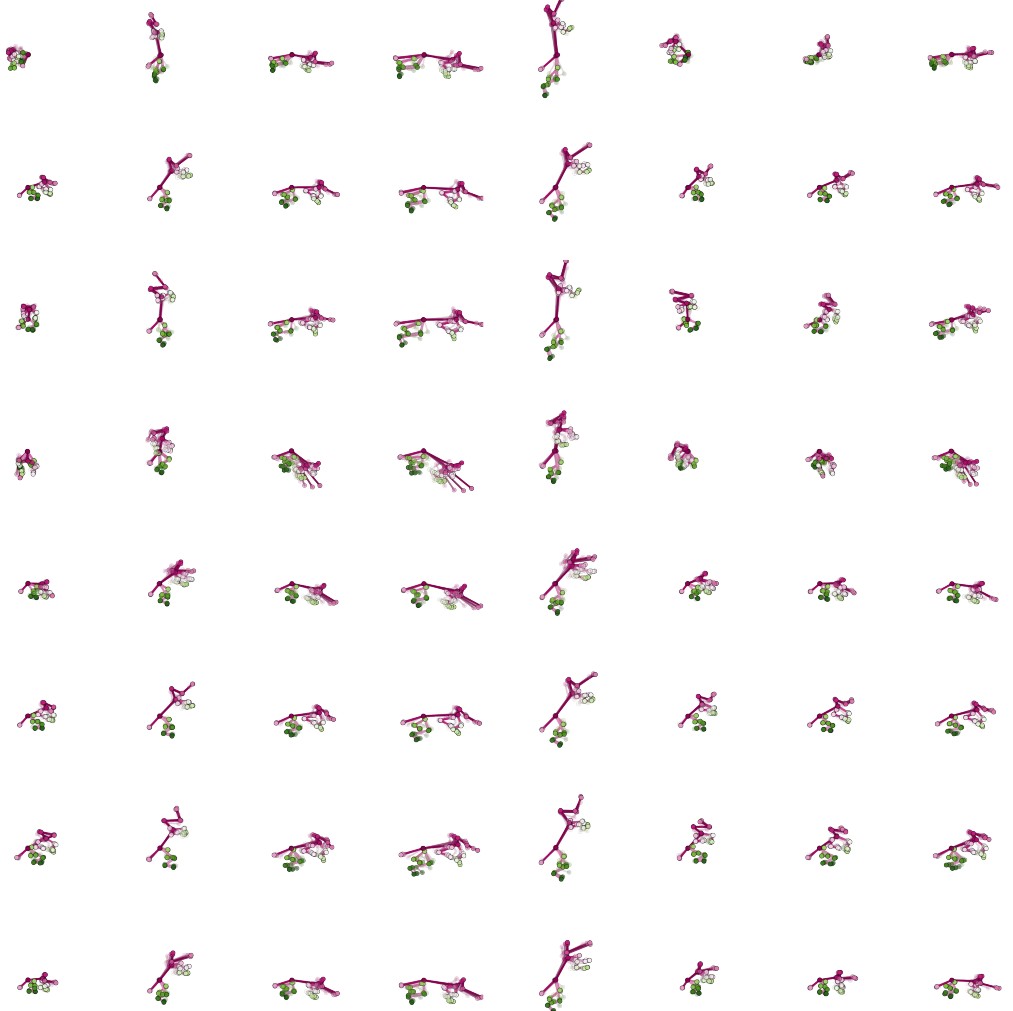

Figure 9: Visualization of the rat-mouse joint embedding space decoded as mice. For the limitation of space, only the first eight bottom codes are shown.

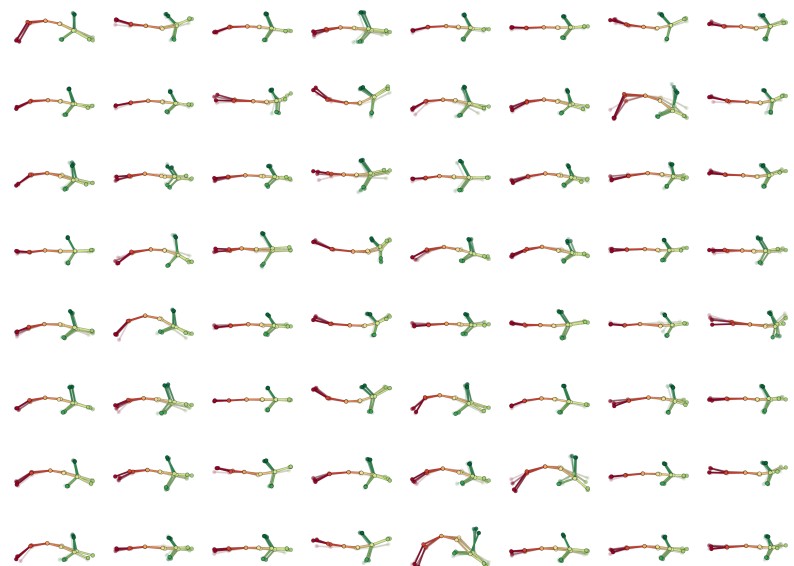

Figure 10: Visualization of the KPMS-2D-CalMS21 joint embedding space for KPMS-2D.

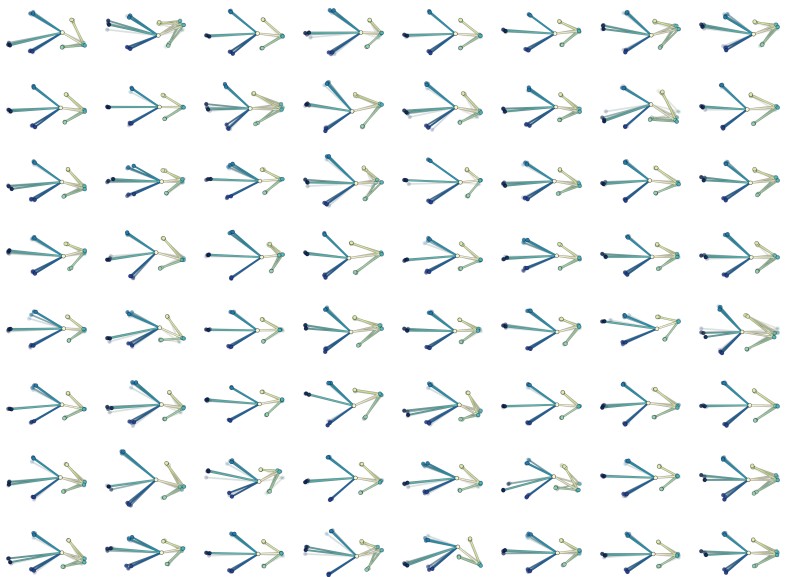

Figure 11: Visualization of the KPMS-2D-CalMS21 joint embedding space for CalMS21.

