# OpenReview forum: "Vector Quantized Representations for Efficient Hierarchical Delineation of Behavioral Repertoires"
_ICLR.cc/2024/Conference — Submitted to ICLR 2024_

### Official Review · Reviewer_pCfH · 2023-10-28

**Soundness:** 2 fair
**Presentation:** 2 fair
**Contribution:** 2 fair
**Rating:** 5
**Confidence:** 2

**Summary:**

The paper presents an efficient framework for dissecting animal behavioral data into hierarchically organized, discrete representations using vector quantization (VQ). The authors demonstrate the effectiveness of their proposed method on real animal body movement analysis and cross-species behavioral mapping tasks.

**Strengths:**

The paper introduces a novel method for analyzing animal behavior, which leverages vector quantization and hierarchical encoding. And they have run experiments on multiple real datasets to analyze animal behavior and map behavior sequences cross-specices.

**Weaknesses:**

The paper lacks more quantitative comparisons. And there are some unclear parts in the paper. I listed questions in the section below.

**Questions:**

- in equation (3), can you elaborate more on the embedding and commitment terms? intuitively what do they mean and how are they derived?

- How robust is the method to variations in the number and granularity of discrete codes? In practice, how do you determine the number of codes in each level?

- what are the quantitative comparison results aginst the KPMS benchmark?

---

> ### Author Response · Authors · 2023-11-22
> **Specific responses to reviewer pCfH**
>
> We thank the reviewer for appreciating the novelty of our proposed method and applications. For your comments on:
>
> - **in equation (3), can you elaborate more on the embedding and commitment terms? intuitively what do they mean and how are they derived?**
>
> Great question and we are happy to elaborate on this. Given the non-differentiability of nearest-neighbor lookup operation from the embedding space, the gradient is directly copied from decoder inputs z_q after quantization to the encoder outputs z(x) (straight-through estimate). A pseudo PyTorch code for this process is given by
> `z_q = z + (z_q - z).detach()`
>
> In this way, the reconstruction loss (first term in equation) can only update the decoder and encoder, not embeddings in the codebook with no gradients passing through. We then included the second term, embedding loss, to optimize the embedding vectors toward the encoder outputs z(x) ia L2 loss. In practice, this loss was replaced by exponential moving average (EMA) to update the codebook embeddings as moving averages of encoder outputs, which provided better performance empirically.
>
> The last term, commitment loss, can be intuitively understood as strategies constraining the volume of the embedding space and encouraging the encoder outputs to ‘commit’ to a specific embedding vector rather than growing arbitrarily, as not controlled by the reconstruction term alone. This also ensures that our straight-through gradient estimates remain concrete.
>
> To summarize, the first reconstruction term optimizes the encoder and decoder; the second embedding term (implemented by EMA) optimizes the embedding vectors in the codebook; the last commitment term optimizes the encoder only. The commitment term is scaled by a hyperparameter alpha in the final objective.
>
> - **How robust is the method to variations in the number and granularity of discrete codes? In practice, how do you determine the number of codes in each level?**
>
> Thank you for the thoughtful comments. We have included new experiments in Table. 1 that examines the changes in downstream performance on motion synthesis with different code granularities. Setting the downsampling rate as 16 frames (320 ms) has yielded overall the best synthesis fidelity and diversity, which is consistent with previous works about the average duration of rodent behaviors (Wiltschko et al. 2015, Marshall et al. 2021). In addition, we have included experiments varying the number of top- and bottom- codes in Supp. Table. 4 in the updated manuscript (also in response to all reviewers).
>
> The number of codes in each level are hyperparameters to tune, but we do deliberately set the number of codes in the top level to be smaller than that in the bottom level, similar to Singh et al. 2019, encouraging disentanglement in the hierarchy. Although it is challenging to form a systematic procedure for tuning the number of codes, as the plausible number of behavioral motifs is more data-driven, we monitored the changes in codebook perplexity (the fraction of inactive codes) during training and examined whether motion sequences derived from the learned motifs are plausible post training.
>
> - **What are the quantitative comparison results against the KPMS benchmark?**
>
> We agree with this comment and have included additional quantitative analyses that compare our method with KPMS (please refer to the overall response to all reviewers). We did not compare with KPMS on the downstream task of motion synthesis, as their model is not designated for such tasks.
>
> We would like to point out that comprehensive, quantitative comparison between behavioral analysis methods has been challenging due to lack of consensus over behavior definitions and benchmarks. To the best of our knowledge, there exists no public continuously annotated rodent behavior datasets (notice the CalMS21 Dataset we have used in the paper is a 2D mouse dataset with annotations on dyadic social behaviors alone). Moreover, the inter-labeler behavior annotation agreement remains low even among well-trained human experts (Segalin et al. 2021) and these labels are inherently biased by human limited understanding of rodent behaviors.
>
> References:
> - Singh et al (2019). Finegan: Unsupervised hierarchical disentanglement for fine-grained object generation and discovery.
> - Wiltschko et al (2015). Mapping sub-second structure in mouse behavior.
> - Marshall et al. (2021). Continuous whole-body 3D kinematic recordings across the rodent behavioral repertoire.
> - Segalinet al. (2021). The Mouse Action Recognition System (MARS) software pipeline for automated analysis of social behaviors in mice.

---

### Official Review · Reviewer_sPmp · 2023-11-01

**Soundness:** 2 fair
**Presentation:** 3 good
**Contribution:** 2 fair
**Rating:** 3
**Confidence:** 3

**Summary:**

This paper describes a framework to learn representations of animal behavioural data, using a VQ-VAE with multi-level encoding. This choice of latent representation enables the data to be decomposed into different discrete behavioural motifs, and enables analysis and synthesis of movements. The application is intriguing, and the presentation is relatively clear.

**Strengths:**

- The theoretical development is sound.
- The presentation is relatively clear.
- The application to animal behavioural data is interesting and important.

**Weaknesses:**

- Unfortunately, it’s not clear whether there is sufficient technical novelty for the ICLR community. The application of VQ-VAE to animal behavioural data specifically may be novel, but other than the multi-level encoding, it is not clear whether there are architectural/learning improvements that may be relevant to the broader ICLR reader. The listed contributions indicate potential novelty in behavioural neuroscience (ie. that this model simplifies the behavioural pipeline), which could perhaps suggest a different venue might be a better fit.
- There is quite a lot of work on existing work on learning movement primitives or behavioural decomposition via latent variable models in embodied and robotics domains. The work could better be positioned in this context, and architectural design choices could be better justified.
- The analysis could be more thorough; as it stands there is only one table of quantitative results, for motion synthesis, and the method is compared mostly to quite weak baselines (fully-connected MLP, GRU).

**Questions:**

- There is quite a lot of work on learning movement primitives (eg. Paraschos et al, 2013), or behavioural decomposition via latent variable models in embodied and robotics domains.
Merel et al (2019a); Bohez et al (2022) apply hierarchical latent variable models to motion-capture data from humans and other mammals, and Merel et al (2019b) specifically studies learned representations of simulated rodent behaviour.
Other works leverage latent variable representations of offline behavioural data in robotics (eg. Singh et al, 2021), including hierarchical discrete representations that can decompose data into discrete motifs that can execute / synthesize meaningful behaviours (Rao et al, 2022).
- Related to these points, it’s not clear why specific architectural choices were made. For example, encoding entire trajectories into a single latent code scales poorly with dimensionality of the inputs and length of the sequence, and many of the approaches from my previous comment use sequential latent variable models to better model embodied temporal data.

Some minor comments:
- Broken citation reference in the first paragraph
- Having the related work as a final section reads a bit awkwardly to me, as it feels like an afterthought. Consider moving it to at least before the final discussion / conclusions, and ideally before the method itself to provide some scaffolding and context for the contributions and claims.

References:
- Paraschos et al (2013), Probabilistic Movement Primitives
- Merel et al (2019a), Neural Probabilistic Motor Primitives for Humanoid Control
- Merel et al (2019b), Deep Neuroethology of a Virtual Rodent
- Singh et al (2021), Parrot: Data-Driven behavioral priors for reinforcement learning
- Bohez et al (2022), Imitate and Repurpose: Learning Reusable Robot Movement Skills From Human and Animal Behaviors
- Rao et al (2022), Learning Transferable Motor Skills with Hierarchical Latent Mixture Policies

---

> ### Author Response · Authors · 2023-11-22
> **Specific responses to reviewer sPmp (1/2)**
>
> We appreciate the reviewer’s constructive feedback for better positioning the motivations and contributions of the proposed method, as well as strengthening the paper with more comprehensive quantitative comparisons. Regarding your comments on:
>
> - **Unfortunately, it’s not clear whether there is sufficient technical novelty for the ICLR community ...**
>
> Thank you for the constructive criticism, and we agree that it is important to ensure that our work is a good fit for ICLR and we believe it is. To your point about how the novelty for neuroscience suggests that a different venue could be a better fit:
> - ICLR has always been a venue for papers on neuroscience applications (also explicitly listed as a subfield in the ICLR Call for Papers), for example, Eyjolfsdottir et al. (2017) for learning recurrent representations for animal behaviors, Li et al. (2022) for animal pose estimation and Molano-Mazon M et al. (2018) for neural activity synthesis, among many more. Papers specifically targeting improving and benchmarking animal behavioral analyses, as we do, have also appeared in other AI/ML conferences including ICML, for instance, Sun et al. (2023) that introduced a multi-species benchmark for learned animal behavioral representations, and NeurIPS, such as Costacurta et al. (2022), Shi et al. (2021) and Batty et al. (2019), to only mention a few. In particular, the methodology presented in Costacurta et al. is a modification to (Keypoint-)MoSeq, the method we have compared to in the paper.
> - Our approach is broadly relevant to neuroscience, extending beyond the scope of behavioral neuroscience. The ultimate goal of neuroscience is to understand how the brain generates behavior, which is, after all, the brain’s principal output. Until recently, analyzing behavior was just not as tractable as analyzing neural activity, as we lacked methods for rigorously and efficiently quantifying the behavior itself. This has changed, and now behavioral analysis methods, and ultimately computational models that relate neural activity to behavior, are key areas of interest and development in computational neuroscience.
>
> To your point about whether the architectural/learning improvements are relevant to the broader ICLR reader:
> - First, please see our above points regarding neuroscience applications in ICLR; at a minimum, the neuro community at ICLR will appreciate this work.
> - Second, despite our focus on applications to behavior and computational neuroscience, our approach is novel at learning the inherent hierarchy among motion primitives using multi-level encoding with a relatively simple, computationally economic framework. This approach could be applicable to either type of sequential data. The state-of-the-art performance on motion synthesis relative to recent methods in human motion synthesis also suggest the benefits for deploying such discrete representations in generative tasks. These tasks are not limited to forecasting or generation of animal movements, but also establishing mappings with other behavioral modalities, such as vocal activities in rodents (languages/speech for animals).
>
> - **The analysis could be more thorough; as it stands there is only one table of quantitative results, for motion synthesis, and the method is compared mostly to quite weak baselines (fully-connected MLP, GRU).**
>
> We appreciate and agree with the constructive feedback and have therefore added quantitative comparisons with Keypoint-MoSeq (please refer to the response to all reviewers). For the task of motion synthesis, we have included experiments with more recent and stronger baselines, for instance MotionDiffuse (Zhang et al. 2022), as well as internal comparisons on our proposed method with different codebook sizes and temporal granularities. We hope that your concerns have been addressed in this revision. We would also like to reiterate that motion synthesis is not the primary focus of this paper, but applications tangential to the analysis of animal behaviors.

---

> > ### Author Response · Authors · 2023-11-22
> > **Specific responses to reviewer sPmp (2/2)**
> >
> > - **(overlapping with the weakness) There is quite a lot of work on learning movement primitives (eg. Paraschos et al, 2013), or behavioural decomposition via latent variable models in embodied and robotics domains …**
> >
> > Thank you for suggesting these relevant references. We agree that these methods are motivating, particularly Merel et al. (2019b) that describes behavioral representations of an artificial rodent model by applying classic analysis methods in neuroscience and Rao et al. 2022 that does offline learning of hierarchical, discrete motor skills. While we acknowledge that our knowledge in RL-oriented skill learning may be limited and biased, the choice of discrete representations versus encoding motion sequences into continuous latent dynamics is better aligned with the longstanding goal of constructing formal taxonomical structures for better describing and understanding animal behaviors (similar ideas also explored in a more RL-oriented setting such as Paulius et al. 2020). The coding scheme we have deployed would better support real-world investigations into behavioral patterns and perturbations, as well as causal relationships across behaviors of different individuals in an information theoretic sense (Ferrer‐i‐Cancho R et al. 2013). Moreover, our studies on naturalistic behaviors in animals are largely disjoint from the goal-directed manipulation of body movements in most RL optimization settings and many findings in related RL literature may not be readily applicable to behavioral analyses. While it would be valuable to offer a comprehensive review of behavioral understanding and analysis in both real organisms and embodied agents, elaborating on this is likely beyond the scope of this paper and deviates from our major focus.
> >
> > - **Related to these points, it’s not clear why specific architectural choices were made. For example, encoding entire trajectories into a single latent code scales poorly with dimensionality of the inputs and length of the sequence, and many of the approaches from my previous comment use sequential latent variable models to better model embodied temporal data.**
> >
> > We appreciate the feedback and apologize if there is any miscommunication with respect to how the proposed model encodes trajectories. The latent variables are still sequential, while a long trajectory is encoded as a long sequence of latent codes. We updated Figure 1 to make this more clear.  Due to the downsampling that occurs from input sequences to latent code, each latent code captures approximately 320ms of the trajectory (out of ~30 minute recordings).
> >
> > - **Broken citation reference in the first paragraph**
> >
> > Thank you for catching this and we have fixed it.
> >
> > - **Having the related work as a final section reads a bit awkwardly to me, as it feels like an afterthought. Consider moving it to at least before the final discussion / conclusions, and ideally before the method itself to provide some scaffolding and context for the contributions and claims.**
> >
> > We agree that it would be helpful to rearrange the Related Work section and thus provide more of an orientation with respect to the motivations and contributions. Please refer to the updated manuscript for changes that have been made.
> >
> > References:
> > - Li et al. (2022). Decompose to Generalize: Species-Generalized Animal Pose Estimation.
> > - Eyjolfsdottir et al. (2016). Learning Recurrent Representations for Hierarchical Behavior Modeling.
> > - Molano-Mazon M et al. (2018). Synthesizing realistic neural population activity patterns using Generative Adversarial Networks.
> > - Batty et al. (2019). BehaveNet: nonlinear embedding and Bayesian neural decoding of behavioral videos.
> > - Costacurta et al. (2022). Distinguishing discrete and continuous behavioral variability using warped autoregressive HMMs.
> > - Shi et al. (2021). Learning disentangled behavior embeddings.
> > - Sun JJ et al. (2023). MABe22: A Multi-Species Multi-Task Benchmark for Learned Representations of Behavior.
> > - Paulius et al. (2020). A motion taxonomy for manipulation embedding.
> > - Ferrer‐i‐Cancho R et al. (2013). Compression as a universal principle of animal behavior.

---

### Official Review · Reviewer_1QG3 · 2023-11-01

**Soundness:** 3 good
**Presentation:** 3 good
**Contribution:** 3 good
**Rating:** 5
**Confidence:** 3

**Summary:**

The authors propose an end-to-end unsupervised behavioral mapping approach that identifies hierarchically organized discrete behavioral motifs from pose time-series data. This is done using a variational encoder to map postural dynamics to a finite-sized discrete embedding with vector quantization.

**Strengths:**

Well written and technical details are clear.  The evaluations are clear.  The motivations are mostly clear and the applications are well explained.

**Weaknesses:**

Missing/failed citation in first paragraph

How is the quantization "codebook" initialized and updated?  This is not clear to me.

An ablation showing the benefit of using the proposed quantization would be helpful.

Were there no other SOTA models to compare against?  The evaluations seem a bit lacking.  Additional applications, comparison models and a detailed ablation would help here as well as more detail on limitations and failure cases.

**Questions:**

How is the quantization "codebook" initialized and updated?

What is the impact of not using quantization on the proposed applications?  This is not clear to me.

---

> ### Author Response · Authors · 2023-11-22
> **Specific responses to reviewer 1QG3 (1/2)**
>
> Thank you very much for the valuable feedback that has helped us to improve the manuscript. Regarding the comments on:
>
> - **Missing/failed citation in first paragraph**
>
> Thank you for pointing this out and we apologize for the error. We have fixed it in the latest version.
>
> - **How is the quantization "codebook" initialized and updated? This is not clear to me.**
>
> Prior to training, the codebook is initialized with zeros. During training, as we had mentioned on Method page 3 in the first version of manuscript, the codebook is updated with exponential moving average (EMA), where each embedding vector is updated as the moving average of encoder outputs. This protocol was first described in the original Van Den Oord et al. VQ-VAE paper and further investigated by Zhang et al. 2023.
>
> - **An ablation showing the benefit of using the proposed quantization would be helpful.**
>
> We agree, thank you for the suggestion. While quantization has the clear benefit of directly inferring the discrete behavioral representations currently used by the computational neuroscience field, we agree that it is important to objectively quantify these benefits. Our previous motion synthesis results actually spoke a bit to this – all external methods we have compared to use a continuous latent representation. But in the revision we include a new comparison to a hierarchical continuous VAE (in the behavioral segmentation tables, see our overall response). We have also attempted a hierarchical, continuous VAE (same architecture but replacing quantization with a standard Gaussian VAE bottleneck, results not shown) but found it challenging to clearly separate different behaviors from post hoc clustering of sample latents, resulting in high perplexity but extremely low average code usage. In this context, quantization plays a role in better structuring the learned latent space to be more interpretable. We elaborate the discussion here in the related question below.
>
> - **Were there no other SOTA models to compare against? The evaluations seem a bit lacking. Additional applications, comparison models and a detailed ablation would help here as well as more detail on limitations and failure cases.**
>
> We appreciate the thoughtful feedback and agree that it would be helpful to strengthen the results by more quantitative comparisons. For motion synthesis, we have now added comparisons with
> - A diffusion-based framework MotionDiffuse (Zhang et al. 2022).
> - Action2Motion (Guo et al. 2020), in addition to the existing GRU VAE baseline.
>
> and internal comparisons with
> - Reconstructed motion sequences by VQ-VAE.
> - Synthesized motion sequences with different downsampling rates.
>
> Please refer to the updated manuscript Table. 1 for detailed results.

---

> > ### Author Response · Authors · 2023-11-22
> > **Specific responses to reviewer 1QG3 (2/2)**
> >
> > - **What is the impact of not using quantization on the proposed applications? This is not clear to me.**
> >
> > To continue with the discussion above, at a high level, the major motivation for the proposed methodology lies in simplifying the pipeline for behavioral analyses, along with making it more efficient and human-interpretable. So, even if quantization merely achieved similar performance on behavioral analysis tasks compared to non-quantized approaches, quantization would still be a major improvement. For example, as we described in Supplement. A4, to obtain ground truth behavior labels out of clustering outputs, multiple human experts were required to independently examine and manually annotate behaviors with textual descriptions, followed by further review to resolve any disagreement, and finally all identified behaviors needed to be grouped into coarser classes for more interpretable delineation of the behavioral repertoire.
> >
> > A continuous parameterization (e.g. Gaussian in most VAEs) results in an embedding space not readily interpretable with respect to the underlying behaviors, and discretization would require post hoc processing by clustering, which can be challenging (Lim et al. 2020). For behavioral analyses, HMMs have also been used for post hoc segmentation of VAE latents into discrete motifs (Luxem et al. 2022), although again requires further human review, annotation, and refinement. Our approach arrives in a single step at the end result of a currently multi-step process, likely at least an order of magnitude improvement in required human hours of effort.
> > However, quantization also improves motion synthesis performance, not just workflow efficiency (which is more challenging to quantify). Our new experiments clearly reveal that quantization improves the downstream motion synthesis task, as quantized codes paired with a lightweight 4-layer autoregressive transformer model (the architecture choice is arbitrary here) outperformed recent SOTA methods that directly operate on continuous sequences.
> >
> > Lastly, we think that it is worth considering how the advantages of quantization have been addressed in previous work on image synthesis (e.g., VQ-VAE2, DALL-E) and speech representation learning (wav2vec 2.0, Baevski et al. 2020). In particular, wav2vec 2.0 illustrated how quantization results in robust training in the presence of noise and artifacts (“... Continuous targets reduce the effectiveness of self-supervised training since targets can capture detailed artifacts of the current sequence, e.g. speaker and background information, which make the task easier and prevent the model from learning general representations beneficial to speech recognition …”). Our quantization scheme could help combat tracking noise and errors as well as body variations across experiment subjects present in animal motion capture data, which we will validate in future experiments.
> >
> > References:
> > - Zhang et al. (2023). Generating Human Motion From Textual Descriptions With Discrete Representations.
> > - Baevski et al. (2020). wav2vec 2.0: A framework for self-supervised learning of speech representations.
> > - Ramesh et al. (2021). Zero-shot text-to-image generation.
> > - Lim et al. (2020). Deep clustering with variational autoencoder.
> > - Luxem et al. (2022). Identifying behavioral structure from deep variational embeddings of animal motion.
> > - Guo et al. (2020). Action2motion: Conditioned generation of 3d human motions.
> > - Van den Oord et al. (2017). Neural discrete representation learning.

---

### Official Review · Reviewer_wDjx · 2023-11-06

**Soundness:** 2 fair
**Presentation:** 3 good
**Contribution:** 2 fair
**Rating:** 5
**Confidence:** 3

**Summary:**

In this paper, the authors propose an end-to-end behavioral analysis approach that dissects continuous body movements into sequences of discrete latent variables using vector quantization (VQ). The discrete latent space naturally defines an interpretable deep behavioral repertoire composed of hierarchically organized behavioral motifs. Using recordings of freely moving rodents, the authors demonstrate that the proposed framework faithfully supports standard behavioral analysis tasks and enables a series of new applications stemming from the discrete information bottleneck, including realistic synthesis of animal body movements and cross-species behavioral mapping.

**Strengths:**

1. The paper is generally well-written and easy to follow.
2. The experimental results seem to support the authors' claims.

**Weaknesses:**

1. It would be better to compare the proposed method with more advanced baseline approaches to demonstrate its effectiveness. There should also be more ablative analysis the illustrate the effectiveness of each component of the model.
2. There is a missing citation on the first page.
3. The major innovations seem not very clear. It would be better to clearly state the major novelty of the proposed method and indicate its advantages over existing methods in the literature. The related work section is suggested to be refined and moved to an earlier place for readers to understand the context of the field.

**Questions:**

Please focus on addressing the issues in the Weaknesses section.

---

> ### Author Response · Authors · 2023-11-22
> **Specific responses to reviewer wDjx**
>
> We sincerely thank the reviewer for the thoughtfulness concerning the assessment of the effectiveness of the work. For the questions listed in the weakness section,
>
> - **It would be better to compare the proposed method with more advanced baseline approaches to demonstrate its effectiveness.**
>
> We appreciate the feedback and have now significantly expanded our comparisons. In addition to new quantitative comparisons for behavioral representations/segmentation, we now compare to a larger set of recent methods for motion synthesis, including ACTOR, Action2Motion and MotionDiffusion (Table 1 in revised text). Notably, our method outperforms even the strong MotionDiffuse baseline, among others. While beyond the scope of this paper, in the future it will be interesting to explore whether these VQ advantages extend to human motion synthesis benchmarks.
>
> - **There should also be more ablative analysis that illustrate the effectiveness of each component of the model.**
>
> Agreed. We have now performed ablative analysis for both behavioral segmentation and motion syntheses, testing the impact of the hierarchical latents, quantization, and codebook size. Details are included in Supp. Table 2 and 3 (same as reported in ‘response to all reviewers’) and Table 1 in the updated manuscript. Notice that we have also attempted a hierarchical, continuous VAE (same architecture but replacing quantization with a standard Gaussian VAE bottleneck, results not shown) but found it challenging to clearly separate different behaviors from post hoc clustering of sample latents, resulting in high perplexity but extremely low average code usage. In this context, quantization plays a role in better structuring the learned latent space to be more interpretable.
>
> - **There is a missing citation on the first page.**
>
> Thank you for pointing this out; have fixed it.
>
> - **The major innovations seem not very clear. It would be better to clearly state the major novelty of the proposed method and indicate its advantages over existing methods in the literature. The related work section is suggested to be refined and moved to an earlier place for readers to understand the context of the field.**
>
> We agree with the reviewer that the major novelty and advantages of the proposed method would be better conveyed to the target audience if the Related Work section is positioned earlier and modified accordingly. Corresponding changes have been updated in the current manuscript, with additional modifications in the Introduction section for better readability.
>
> Our approach is capable of delineating animal behavioral repertoire from motion capture data, in a more computationally efficient and human-interpretable way than established methods and is novel at finding the inherent type/subtype hierarchy among motion primitives without post hoc manual annotation and grouping. . The hierarchical representations recapitulate the standard behavioral analysis tasks as existing techniques, while quantitatively capturing and grouping finer-scale kinematic details, as well as enabling novel tasks.

---

### Author Response · Authors · 2023-11-22
**General responses to all reviewers**

We sincerely appreciate the reviewers’ thoughtful suggestions and comments regarding our submission. The majority of the comments focused on strengthening the paper with additional quantitative comparisons and ablation studies, with an emphasis on  behavioral segmentation (comparing to Keypoint-MoSeq) and motion synthesis (stronger and more recent baselines). Most reviewers also suggested that we contextualize our method’s contributions earlier in the paper for the ICLR audience. In addition to directly responding to each reviewer’s comments, we have:
performed additional quantitative analysis, while comparing our method with Keypoint-MoSeq (KPMS, the method we made qualitative comparisons to in the original submission).
implemented additional baselines and internal comparisons for the performance of motion synthesis, as updated in Table. 1.
polished the text and figures. As requested, we also moved the Related Work section to follow the Introduction, and we now provide a more thorough orientation of major motivations and contributions and key differences from past work.

**Quantitative experiments for comparing our method with KPMS:**
- **Perplexity**: a common information metric used for sequence analyses (e.g., language models), given by $PP(x) = 2^{H(x)}$, where $H(x)$ is the entropy of code usage. For codebook/motif learning, lower perplexity implies possibilities for codebook collapse, i.e., a smaller number of active embedding vectors in the codebook. Higher perplexity thus implies more diverse sampling of the behavioral repertoire, i.e. that fine-grained differences in behaviors are detected and assigned to separate codes rather than being subsumed by a smaller number of coarse codes. For these comparisons, we fixed the codebook size (for quantization) and number of KPMS hidden states (n = 128). For KPMS, we also tested multiple $\kappa$ parameter values, which controls state transition frequency.

|                       | KPMS             | KPMS             | KPMS             | VQ-VAE         | Ours          | Ours          |
|-----------------------|------------------|------------------|------------------|----------------|---------------|---------------|
|                       | ($\kappa =10^3$) | ($\kappa =10^5$) | ($\kappa =10^7$) | (no hierarchy) | (TC=4, BC=32) | (TC=8, BC=16) |
| Perplexity $\uparrow$ | 16.63            | 15.17            | 14.94            | 56.23          | 57.38         | 57.87      |
| $\geq 1$\% usage      | 15.63\%          | 13.28\%          | 12.5\%           | 17.19\%        | 21.88\%       | 23.44\%    |

Despite tuning of the hyperparameter $\kappa$, KPMS yielded perplexity significantly lower than both a non-hierarchical and hierarchical VQ-VAEs, indicating that these VQ methods better capture fine-grained behaviors. At the same time, our hierarchical method not only captures these fine-grained behaviors but also groups them automatically into coarser categories, offering multiple scales of description in one shot.

- **Jensen Shannon Divergence** for behavioral subtypes: We computed distributions of code/syllable usage for different subtypes under the same coarse behavior category (e.g., high rearing versus low rearing) and assess the method’s capacity for distinguishing subtypes via JS divergence (the higher the better).

Our method is more capable of detecting subtle differences in kinematics than KPMS, especially in behavioral subtypes with salient postural changes. Even with only 8 and 16 codes, the performance of our method is on par with the non-hierarchical VQ-VAE, which contains 128 distinct codes.

| Coarse Class          | Rearing            | Idle              | Locomotion |
|-----------------------|--------------------|-------------------|------------|
| Subtypes              | High Rear-Low Rear | Head Up-Head Down | Slow-Fast  |
| KPMS ($\kappa =10^3$) | 0.389              | 0.222             | 0.213      |
| KPMS ($\kappa =10^5$) | 0.360              | 0.175             | 0.219      |
| KPMS ($\kappa =10^7$) | 0.303              | 0.155             | 0.282      |
| VQ-VAE (no hierarchy) | 0.748              | 0.606             | 0.356      |
| Ours (TC=4, BC=32)    | 0.611              | 0.557             | 0.370      |
| Ours (TC=8, BC=16)    | 0.610              | 0.561             | 0.340      |
| Ours (TC=4, BC=16)    | 0.578              | 0.435             | 0.325      |
| Ours (TC=8, BC=12)    | 0.631              | 0.545             | 0.278      |
| Ours (TC=8, BC=32)    | 0.700              | 0.580             | 0.366      |

---

### Meta-Review · Area_Chair_49Fo · 2023-12-09

**Metareview:**

Authors made many substantial improvements to the manuscript in response to reviewer concerns, but a critical gap still exists.

The primary issue is a disconnect between the expectations set in the manuscript and the results provided. Reviewers were left with the impression, and rightfully so, that the main contribution is a method to improve the analysis of rodent motion datasets. The manuscript describes a method to do so, but it never delivers its punchline: actually showing how its methods support a deeper analysis of rodent motion datasets. Instead, it shows better motion generation, better alignment to human behaviors, and argues that its method better captures rodent motion. This is the underlying cause for why reviewers felt the manuscript doesn't quite fit the venue, it's lacking what should be its main contribution.

This can also be seen in the responses to reviewers: "... the major motivation for the proposed methodology lies in simplifying the pipeline for behavioral analyses, along with making it more efficient and human-interpretable. So, even if quantization merely achieved similar performance on behavioral analysis tasks compared to non-quantized approaches, quantization would still be a major improvement." But the utility of this key idea is never demonstrated in the manuscript.


The authors add numerous results in response to reviewer requests. Including comparison with state-of-the-art methods. Yet, while these are produced, they are not contextualized and cannot be understood in their current form. For example, Table 1 in the manuscript, a new addition, is never referenced, has a short and very ambiguous description, and provides no added comment for these results. One might wonder why prior methods are far worse when applied to these datasets than they are in the original submissions. We appreciate that authors added SOTA comparisons, and this will undoubtedly strengthen their next submission, but this is a significant amount of added material to review (amounting to roughly as many results as the original submission had) which must first be more fully incorporated and explained by the authors.

This could be a good submission, particularly with the added pieces suggested by reviewers, but requires more polish, integration, and explanation. I also encourage the authors to more directly tackle their target domain.

**Justification For Why Not Higher Score:**

Adding so many new results means that the manuscript requires a thorough revision.

**Justification For Why Not Lower Score:**

N/A

---

### Decision · Program_Chairs · 2024-01-16

Reject